# The Governance of Childhood Vaccination Services in Crisis Settings: A Scoping Review

**DOI:** 10.3390/vaccines11121853

**Published:** 2023-12-14

**Authors:** Nada Abdelmagid, Rosamund J. Southgate, Mervat Alhaffar, Matab Ahmed, Hind Bani, Sandra Mounier-Jack, Maysoon Dahab, Francesco Checchi, Majdi M. Sabahelzain, Barni Nor, Bhargavi Rao, Neha S. Singh

**Affiliations:** 1Department of Infectious Disease Epidemiology and International Health, Faculty of Epidemiology and Population Health, The London School of Hygiene & Tropical Medicine, London WC1E 7HT, UK; 2Health in Humanitarian Crises Centre, The London School of Hygiene & Tropical Medicine, London WC1E 7HT, UK; 3Independent Consultant in Public Health, Oxfordshire OX4 4ER, UK; 4Syria Research Group (SYRG), Co-Hosted by the London School of Hygiene and Tropical Medicine, London WC1E 7HT, UK and Saw Swee Hock School of Public Health, National University of Singapore, Singapore 117549, Singapore; 5School of Health Sciences, Ahfad University for Women (AUW), Omdurman P.O. Box 167, Sudan; 6Department of Global Health and Development, Faculty of Public Health and Policy, The London School of Hygiene & Tropical Medicine, London WC1H 9SH, UK; 7School of Public Health, Faculty of Medicine and Health, University of Sydney, Sydney, NSW 2050, Australia; 8Department of Women’s and Children’s Health, Uppsala University, 751 23 Uppsala, Sweden

**Keywords:** vaccination, governance, humanitarian, scoping review

## Abstract

The persistence of inadequate vaccination in crisis-affected settings raises concerns about decision making regarding vaccine selection, timing, location, and recipients. This review aims to describe the key features of childhood vaccination intervention design and planning in crisis-affected settings and investigate how the governance of childhood vaccination is defined, understood, and practised. We performed a scoping review of 193 peer-reviewed articles and grey literature on vaccination governance and service design and planning. We focused on 41 crises between 2010 and 2021. Following screening and data extraction, our analysis involved descriptive statistics and applying the governance analysis framework to code text excerpts, employing deductive and inductive approaches. Most documents related to active outbreaks in conflict-affected settings and to the mass delivery of polio, cholera, and measles vaccines. Information on vaccination modalities, target populations, vaccine sources, and funding was limited. We found various interpretations of governance, often implying hierarchical authority and regulation. Analysis of governance arrangements suggests a multi-actor yet fragmented governance structure, with inequitable actor participation, ineffective actor collaboration, and a lack of a shared strategic vision due to competing priorities and accountabilities. Better documentation of vaccination efforts during emergencies, including vaccination decision making, governance, and planning, is needed. We recommend empirical research within decision-making spaces.

## 1. Introduction

A record 274 million people worldwide needed humanitarian assistance in 2023 [1], including 149 million children [2]. Crisis-affected populations are particularly vulnerable to elevated morbidity and mortality from infectious diseases driven by multiple risk factors, including crowded living conditions; forced displacement; poor quality shelter; poor access to safe water, sanitation, and hygiene; inadequate access to health services; and underperforming epidemic surveillance [3]. As a result, vaccine-preventable diseases (VPDs) in crisis-affected settings remain a pressing health concern. Up to 2 million ‘zero-dose’ children (those who have not received their first dose of the diphtheria-tetanus-pertussis (DTP) vaccine) and 3.8 million under-immunised children (those who have not received their third dose of DTP) live in conflict-affected settings [4]. These children are often also deprived of other essential services and face socio-economic and gender disparities that may limit vaccination services access or demand, such as the indirect costs of seeking vaccination or the gendered dimensions of household or family decision making for seeking child health services [5]. This has repercussions for these communities due to heightened risks of outbreaks, avoidable child deaths, poor child development outcomes, and medical impoverishment and poverty [6,7,8]. 

Global improvements in routine immunisation services mask significant inter- and intra-country disparities [9], warranting novel approaches to reach persistent pockets of unvaccinated children, including those in crisis-affected settings. Inadequate control of VPDs in crisis-affected settings poses major challenges to global health security and development goals, such as polio eradication [10], effective epidemic preparedness and response [11], Immunization Agenda 2030 [12], and universal health coverage [13]. Despite recent innovations with the potential to address barriers to vaccine delivery in crisis-affected settings, such as microarray patches, shelf-stable vaccines, and needleless injectors [14,15,16,17], evidence from the last decade suggests that vaccines against Streptococcus pneumoniae, Rotavirus, and Haemophilus influenzae type-b, which cause significant morbidity and mortality, continue to be underutilised in crisis-affected settings, and that vaccination services are frequently untimely or inappropriately delivered [18,19,20]. 

Despite calls for a systematic, evidence-based, and accountable approach to vaccine provision in humanitarian crises [19,21] and the development by the World Health Organization (WHO) of a decision-making framework for vaccination in acute humanitarian emergencies [22], the persistence of inadequate vaccination interventions in crisis settings calls into question the processes of decision making about which vaccines to use, when, where, how, and for whom. With multiple and largely autonomous global, national, and local actors and often limited governmental effectiveness in crisis-affected countries, decision making around vaccination services often lacks structure and transparency, and vaccination services are likely not exempt from the challenges of overall healthcare governance in humanitarian settings [23]. Furthermore, there is insufficient literature concerning the uptake of the WHO decision-making framework; Rull et al. applied it in South Sudan and have expressed reservations about its usability and familiarity with critical stakeholders [24]. The literature also reports that obstacles to effective vaccination decision making include the absence of guidance and contextually relevant research, operational barriers such as security, and political considerations [21]. However, there is insufficient evidence on the nature and features of vaccination governance and decision making in crisis-affected settings.

Recently, the adoption of different strategies to reach zero-dose children and their communities has been put forward as a pathway towards achieving universal health coverage (UHC) (2). For example, in 2022, Gavi, the Vaccine Alliance launched a USD 100 million Zero-Dose Immunization Programme (ZIP) with the International Rescue Committee and World Vision to reach zero-dose children in the Sahel region and the Horn of Africa—two regions that host millions of zero-dose children [25]. Millions of zero-dose children are crisis-affected, so this agenda provides a unique opportunity to improve childhood vaccination in crisis-affected settings. 

To inform the design of such strategies, we reviewed the literature to describe the key features of childhood vaccination service design and planning in crisis-affected settings. We investigated how the governance of childhood vaccination services in crisis-affected settings is defined, understood, and practised. We argue that understanding and strengthening childhood vaccination governance in crisis-affected settings is vital to improve the effectiveness of vaccination interventions in reducing excess morbidity and mortality and to improve the equitable delivery of vaccines to zero-dose and under-immunised communities.

## 2. Materials and Methods

We did a scoping review using Arksey and O’Malley’s six-stage scoping framework [26] with Levac et al.’s 2010 revisions [27] and Khalil et al.’s 2016 refinements [28]. Based on our initial examination of the evidence to inform the review protocol, we observed a lack of literature dedicated to the topic. Consequently, we recognised the need to search for and include a wide range of literature types and sources in our review. The chosen methodology caters explicitly to this situation and supports an iterative team-based approach to screening literature and extracting data. The methodology also guided the alignment of our study aim and research question.

### 2.1. Stage 1. Defining the Research Question 

We specified the following research question: ‘What are the main features of the governance of childhood vaccination service planning in crisis settings?’We used the following working definitions:Crisis settings refer to 41 countries with a United Nations consolidated appeal/humanitarian response plan for at least one year during 2010–2021 (Appendix A).Vaccination refers to any services that aim to deliver antigens, through routine or supplementary modalities, to children under five years of age in crisis settings.Governance refers to the processes of interaction and decision making among the actors involved in vaccinating children that lead to the creation, reinforcement, or reproduction of social norms and institutions [29].Vaccine service planning includes ‘who’ (actors and their roles), ‘what’ (choices of vaccines, vaccine delivery modalities (e.g., routine and mass campaigns), and targeted populations), and ‘how’ (e.g., sourcing of vaccines/cold chain/funds, interventions to increase uptake).

### 2.2. Stage 2. Identifying Relevant Studies

We searched the EMBASE, Global Health, CINAHL Plus, and Web of Science databases for peer-reviewed articles on 5 May 2022. We searched Google, Google Scholar, and ReliefWeb for grey literature reports on 5, 6, and 16 May 2022, respectively. We reviewed the first 100 results of each engine search to capture the most relevant hits.

Search terms covered four concepts: crisis settings, vaccination, governance, and service planning (see Appendix A). We also searched subject headings for vaccination, governance, and service planning.

SMJ and FC recommended three documents, which were included in the screening.

### 2.3. Stage 3. Selecting Studies

The inclusion and exclusion criteria are described in Table 1. Google, Google Scholar, and ReliefWeb search results were bookmarked and screened in the web browser used at the time of searching. Other citations were exported to EndNote (Version 20, Clarivate Analytics, Philadelphia, United States of America) and Zotero (version 6.0.26, Center for History and New Media at George Mason University, Virginia, United States of America) for screening and selection. 

Document selection involved a four-stage process: (i) removing duplicates automatically and manually, (ii) screening titles and abstracts of search results to remove ineligible documents in EndNote (NA, MA, SMJ, and RJS did duplicate screening; NS resolved disagreements on screening decisions), (iii) reviewing the full-text articles of search results to remove ineligible documents in Zotero, and (iv) selecting documents for inclusion. 

### 2.4. Stage 4. Charting Data 

We extracted the following variables from each eligible study into an Excel database: source citation, context (country), nature of crisis, years of vaccination activities; vaccination-related data (vaccines considered/used, vaccination aims, presence or absence of vaccine-preventable disease transmission, vaccination delivery modes and strategies, age and type of population served by vaccination services), vaccination actors and their roles, sources of vaccines and cold chain, sources of funding for vaccines, cold chain and service delivery, cost of vaccines and cold chain, communication interventions for community engagement aims and modes, and integration of other services with vaccination. Where relevant, we extracted verbatim text from any document that mentioned governance, as defined above.

### 2.5. Stage 5. Collating, Analysing and Reporting Results

We analysed the descriptive data extracted into Excel using frequency counts and proportions for each variable. 

We used Dedoose (version 6.0.26, Center for History and New Media at George Mason University, Virginia, United States of America) to extract and analyse text excerpts from eligible documents on governance. We used a deductive and inductive coding approach guided by the governance analysis framework (GAF) [29], explicitly developed for observing and analysing governance processes from a non-normative stance, emphasising description and analysis without reference to a standard or optimal solution. Six deductive root codes were created, including one for ‘definitions of governance’ and five corresponding to the analytical tools of the GAF, namely problems, social norms, actors, nodal points, and processes (Appendix A). ‘Child’ codes were created inductively. NA and RJS conducted coding. NA synthesised coded excerpts and supplemented these with supportive examples.

We report findings in three sections. The first section describes the key features of vaccine service planning for children in crisis settings per the reviewed literature. The second section presents findings on how governance is defined and understood. The third section describes the different governance arrangements (the actors involved, the nature of their influence, the spaces where they interact, and the nature of their interactions, including prevailing norms) and presents findings by four interlinked domains of vaccination: funding for vaccines and vaccination services; access to vaccine stocks; setting vaccination goals and standards; and designing, planning, and providing vaccination services.

### 2.6. Stage 6. Consulting Stakeholders

NA and RJS held a coding validation session with NS, MA, and MD. A synthesis discussion session of initial findings was held by NA with members of the research team who were not involved in this review for their feedback on themes and interpretation of findings. 

## 3. Results

### 3.1. Description of Eligible Documents

We identified 193 eligible documents (Figure 1). Specific references to eligible crises were found 219 times in the documents: 11.4% (*n* = 25) of reports were for Afghanistan, 10% (*n* = 22) for South Sudan, and 9% (*n* = 19) of documents covered crisis-affected settings generally. Among the 219 crisis-specific reports, 65.3% (*n* = 143) were for settings affected by conflict or insecurity, and 9.6% (*n* = 21) were in settings with multiple concurrent crises. The data or information reported in the included documents ranged from 2005 to 2022. More detail is reported in Figure 2 and Figure 3.

Specific vaccines were referenced 297 times. Overall, 64% (*n* = 123) of the included documents reported on one vaccine, 18% (*n* = 34) reported on two, and 14% (*n* = 27) reported on three to six. Polio vaccine was the most commonly reported (30%, *n* = 89), with 18.2% (*n* = 54) mentions of both cholera and measles vaccines and 17.2% (*n* = 51) references to routine childhood vaccines generally. 

Among the 297 vaccine reports, 43.1% (*n* = 128) were linked to ongoing transmission or outbreaks, while 29% (*n* = 86) were focused on epidemic response, and 14.5% aimed at disease elimination (*n* = 43). Mass delivery through campaigns was mentioned in 40.4% (*n* = 120) of cases, while only 27.9% (*n* = 83) specified delivery approaches, often involving a combined approach. The population targeted was not specified in 37% of reports (*n* = 110), with 15.8% (*n* = 47) addressing low-coverage or high-risk groups and 21.9% (*n* = 65) referring to multiple population categories. Child age groups were unspecified in 54% (*n* = 160) of reports, with 20.2% specifying children under five years (*n* = 60). Critical details, such as the sources of vaccines, funding, cost per dose, sources of the cold chain, funding of the cold chain, and cost per dose of the cold chain, were notably absent in almost all instances. More detail is reported in Table 2.

Regarding vaccination actors, 64% of eligible documents (*n* = 69) referred to between three and six unique actors, while 28% of documents (*n* = 55) mentioned two actors, 28% (*n* = 55) mentioned one actor, and 15% (*n* = 29) did not mention any actors. In total, specific actors were mentioned 448 times in eligible documents. The most mentioned actors were UNICEF (17.9%, *n* = 80), the World Health Organization (WHO) at headquarters or regional levels (15.8%, *n* = 71), and national ministries of health (MOH) in crisis-affected countries (13.4%, *n*= 60). Implementation was the most discussed or reported role (12.7%, *n* = 5). In comparison, in 14.1% (*n* = 63) of instances, actors were described as having multiple roles, for example, vaccine procurement and funding for UNICEF and technical support and coordination for WHO. The roles were not specified in 32.1% (*n* = 144) of instances when actors were mentioned. The full features of the 448 actor reports are reported in Figure 4 and Figure 5.

### 3.2. Definitions and Interpretations of Childhood Vaccination Governance

We identified broad-ranging conceptualisations, definitions, and interpretations of the concept of vaccination governance in crisis settings (Table 3).

#### 3.2.1. As Programme Management (*n* = 7)

Some of the literature framed governance in the context of programme management. Governance is described as the systems of authority and accountability that shape and oversee the management and performance of vaccination programs. It encompasses various elements, such as policies for vaccinating refugees and migrants [30], national systems addressing operational immunisation issues [31], program structures that define roles and responsibilities [32,33], strategies for program improvement [34], and guidelines for vaccination activities and adverse events following immunisation (AEFI) reporting systems [35].

#### 3.2.2. As Leadership and Ownership (*n* = 32)

In other literature, governance was understood in the context of leadership or ownership of vaccination interventions. According to the authors, leadership is expressed through various means, including the commitment of senior government officials in crisis-affected countries [36], visible involvement in vaccination campaigns and high-level meetings [33,37,38], financial investments [32,33,39,40,41], intervention in cases of inadequate performance or crises [32], and autonomous decision making related to vaccination [42,43,44]. Leadership is also seen in the coordination of diverse actors in vaccination [45], with national governments or UN agencies, particularly the WHO and UNICEF, often taking the lead [46,47,48,49,50,51,52]. Co-leadership arrangements between governments and UN agencies are common [38,50,53,54,55,56,57,58]. Leadership also involves governments’ ability to gain public cooperation [59,60], declare disease outbreaks [61], and develop comprehensive vaccination strategies [36,47,62,63,64].

#### 3.2.3. As Accountability (*n* = 28)

The literature also associates governance with accountability within the context of vaccination programs. Accountability involves measuring and reporting vaccination outcomes such as vaccination coverage [32,55,56,65,66,67,68,69], VPD outbreak notification [61], or AEFI reporting systems [35], ensuring qualified personnel, justifying activities, and using accountability frameworks. According to the literature, the main indicator of programme performance was the intended or achieved number of children vaccinated or missed children [65,66,70,71,72]. The status of an outbreak after a reactive vaccination intervention was also mentioned as a proxy performance indicator [70]. Accountability was also interpreted as ensuring the availability of qualified personnel [32,33,34,73,74,75], efforts to meet global targets [47,67,76,77,78], using accountability systems and frameworks [32,74,79,80], involving the affected population or vaccination end users [77], and ensuring the timeliness and completeness of vaccination data [81]. Simpson et al. distinguished between financial, legal, and technical accountability concerning childhood vaccination in Afghanistan [38].

#### 3.2.4. As Decision Making (*n* = 18)

In some documents, governance was framed as decision making. The contexts of vaccination decision making were discussed either as centralised versus devolved [47] or in relation to the degree of autonomy a national MOH had to make vaccination-related decisions [43]. Decision making was framed by some authors as a prioritisation exercise, in terms of the populations or geographical areas targeted [43,56,69,82,83,84,85,86,87,88], of vaccination over other public health interventions [56,88,89,90,91,92], or between specific vaccines [24,58,63,92]. None of the papers discussed the inclusion of different stakeholders in decision making.

#### 3.2.5. Other Definitions of Governance (*n* = 2)

Decobert also referred to governance as affording legitimacy to health workers and health systems delivering vaccination services through official recognition [47]. Two papers framed governance in relation to the degree of financial and technical independence of actors to define and achieve vaccination aims, such as their ability to design, resource, manage, and deliver vaccination services independently [47] or the ability of national ministries of health in crisis-affected countries to enhance their institutional capacity and sustain it independently [41].

### 3.3. Actual Governance Arrangements

In childhood vaccination within crisis settings, decision making involves global, regional, national, and sub-national levels across four interconnected domains: funding, access to vaccine stocks, setting standards, and service provision. Governance arrangements in these domains are presented, detailing the actors, their influence, interaction spaces, and norms, highlighting adaptations across contexts.

Broadly, governance arrangements are ambiguous. Overall, we identified a dominance of a few strategic actors in most decision-making spaces, particularly spaces where decisions about funding, vaccine stock allocations, and setting goals and standards are made. Less strategic but relevant actors have some influence on designing and planning the operational delivery of interventions within national or sub-national spaces, and the influence of strategic actors extends to both defining norms and paving the way for norm adaptations.

#### 3.3.1. Governance Arrangements for Funding Vaccines and Vaccination Services (*n* = 19)

Information regarding the funding sources for vaccines and vaccination services is limited. 

Vaccine funding operates through global stockpiles and in-country channels for routine vaccination or epidemic response. The global oral cholera vaccine (OCV) stockpile, available for cholera control [89], is primarily funded by GAVI (56) [43], with contributions from entities like the Bill & Melinda Gates Foundation and the European Union [83]. It is unclear if crisis-affected regions contribute to the OCV stockpile. UNICEF plays a significant role in financing routine childhood vaccines in crisis-affected countries, sometimes with support from sources like UK Aid [93]. Other funders for in-country vaccination responses include the European Civil Protection and Humanitarian Aid Operations (ECHO) [94], WHO [95], sub-national health authorities [95], Médecins Sans Frontières (MSF) [96], and the Measles and Rubella Initiative (MRI) [97]. Funding for operational costs of vaccination services in crisis settings is also available, supported by organisations like GAVI [93] and the Central Emergency Response Fund (CERF) [98].

There was little information on where actors interact and where decisions are made regarding the funding of vaccines and vaccination services. Furthermore, while GAVI’s decisions to support the global OCV stockpile are made solely by the GAVI Board [83], it is unclear if this is informed by inputs from other actors, such as the stockpile’s decision-making partners. 

The review revealed two common features of funding for vaccines and vaccination services: it is often vertical, and donors are hesitant to make long-term financial commitments [99], particularly related to broader health systems [100] and health governance and leadership [31]. Many crisis-affected countries, including Chad [31], Afghanistan [101], and Yemen [102], heavily rely on external donors, notably GAVI and the Global Polio Eradication Initiative (GPEI), to maintain immunisation services. This donor dependency is acknowledged by UN agencies, MOH, NGOs, and communities, but there is an understanding that this support will eventually cease. For example, UNICEF highlights its reliance on costly outreach activities funded by GAVI in Sudan, which is considered unsustainable [103]. In Afghanistan, donor funding is praised for immunisation achievements, yet there is a recognition of the need to explore alternative means of financial sustainability as donor support diminishes [101]. In Somalia, there is widespread concern about the withdrawal of GPEI funding and a lack of confidence in the MOH’s ability to sustain comprehensive vaccination services independently [51].

#### 3.3.2. Governance Arrangements for Access to Vaccine Stocks (*n* = 20)

Information about accessing vaccines for children in crisis-affected settings is moderately available. During outbreaks and crises, vaccines can be sourced from various organisations and mechanisms, including UNICEF, GPEI, MRI, the International Coordinating Group (ICG), the Humanitarian Mechanism, or through direct purchases from manufacturers [104]. The primary focus in eligible documents is on accessing vaccines from stockpiles, notably the global OCV stockpile, with limited information on other sources or direct purchases.

The global OCV stockpile coordinates the emergency deployment of vaccines, with the ICG overseeing the process and the WHO acting as the secretariat [89]. Decision-making partners for the stockpile are the International Federation of Red Cross and Red Crescent Societies (IFRC), UNICEF, WHO, and MSF [43,83,88,105]. Crisis-affected country governments or decision-making partners can submit requests for OCV [65,93,105,106,107]. Still, the approved quantity may be less than requested, as was the case previously in Bangladesh [89], South Sudan [43], and Yemen [105].

Manufacturers control the prices of vaccines when purchased directly from them. Until 2016, pneumococcal vaccines (PCV) were available at commercial prices from Pfizer and GSK. For years, MSF negotiated with Pfizer and GSK and temporarily accepted a donation after failed negotiations in 2014 [83]. In 2016, GSK and Pfizer announced lowered PCV prices as part of their humanitarian assistance efforts [108].

Since its launch in 2017, the emergency deployment of PCV and rotavirus vaccines from the Humanitarian Mechanism has been coordinated and managed by the WHO, UNICEF, MSF, and Save the Children as decision-making partners [109]. There was no information on deployments of PCV from the Humanitarian Mechanism in the eligible documents.

Beyond stockpiles, UNICEF plays a leading role in supporting governments in affected countries in procuring vaccines for VPD outbreak prevention and response, such as in Libya [110], Niger [111], and Ethiopia [63] in 2018. The reviewed documents do not provide explicit information on the sources of vaccines and the costs associated with procurement.

Information on where vaccination actors interact and decide whether and how to access vaccine stocks is limited. Global stockpiles for OCV, PCV, and rotavirus are controlled by IFRC, MSF, UNICEF, WHO, and Save the Children. Requests for OCV stock are generally limited to governments or country offices of decision-making partners in crisis-affected countries. The influential role of stockpile partners is also reflected at the country level, as seen in the integration of OCV into the cholera response in South Sudan with the involvement of MSF [88].

There is a widespread understanding among MOH, UN agencies, and NGOs in crisis-affected countries that vaccine availability is limited [112]. Consequently, decisions often prioritise using limited supplies [83,113]. Availability challenges persist even after the establishment of global stockpiles, as seen in South Sudan and Bangladesh [43,89]. Prioritising limited vaccines in-country can lead to delayed decision making and implementation of vaccination interventions and concerns about social unrest due to non-universal vaccination [43].

Calls have been made for guidance on how and when to use vaccines reactively [43], tools to prioritise populations for maximum health benefits [56], simplified procedures to access vaccines from global stockpiles [43], and the use of locally produced vaccines in countries affected by crises, including vaccines that are not pre-qualified by the WHO, to reduce the gap between global demand and supply [61]. Additionally, there is a shift in attitude away from the prevailing norm that humanitarian organisations and NGOs should purchase vaccines at commercial prices [83]. The establishment of the global OCV stockpile and the Humanitarian Mechanism were also a reflection of a changing global attitude towards improving access to vaccine stocks for responders in crisis-affected countries [43,109], influenced by scientific evidence, prior expertise, stakeholder consensus, WHO guidelines, and a World Health Assembly (WHA) resolution [83,113].

#### 3.3.3. Governance Arrangements for Setting Goals and Standards (*n* = 37)

The review revealed two strategic actors with sufficient power to set relevant vaccination goals and guidelines. The World Health Organization (WHO) is instrumental in prequalifying vaccines [61], issuing technical guidelines [24], and establishing financial mechanisms for global vaccine stockpiles [113]. The Global Polio Eradication Initiative (GPEI) plays a vital role in influencing vaccination efforts in conflict-affected areas where polio eradication is challenging, such as Afghanistan, Pakistan, and Nigeria. The GPEI sets polio eradication targets and develops guidelines for transitioning polio assets to other public health areas [51,59].

These actors interact within specific decision-making spaces at both the global and country levels, and their influence is widespread. At the global level, they include high-level UN bodies, for example, the UN General Assembly [47,114] and the WHA [86,115], GPEI governance structures [51,116,117], and UN-coordinated global platforms [40,72,118]. Global standards and goals profoundly impact country- and regional-level vaccination decisions [38]. For example, WHA polio eradication resolutions, the Polio Eradication and Endgame Strategic Plan, and GPEI’s polio transition planning guidelines have triggered or shaped WHO Africa region, Nigeria, Somalia, South Sudan, and Afghanistan immunisation strategies and plans relevant for crisis-affected populations [51,86,116,117,119]. Similarly, the global Reaching Every District (RED) immunisation strategy, the UHC target, and the Global Measles Elimination Strategy have been influential in vaccination service design and reporting in Uganda [118], Myanmar [47], and South Sudan [72], respectively.

These decision-making spaces tend to involve a limited set of actors, especially national governments, WHO, UNICEF, and select global organizations. This leaves little room for other stakeholders, including crisis-affected populations. Advisory bodies, such as GPEI Technical Advisory Groups, significantly shape vaccination decisions at the country level, as seen in polio planning spaces in Nigeria [86], annual Expanded Programme on Immunization (EPI) meetings in South Sudan [72], and polio transition planning in Somalia [51]. Furthermore, some advisory bodies, such as GPEI Technical Advisory Groups (TAG), can significantly shape vaccination decisions at the country level [38,116].

There are diverse attitudes towards adherence to technical guidance and standards in vaccination, with some valuing guidelines [73] and others open to flexibility [36]. At the same time, some governments seem occasionally reluctant or unable to conform to global norms [120]. These diverse attitudes could be partly attributed to the unique challenges in crisis-affected settings, which limit the ability to meet strict standards. This is exemplified by the increasing use of single-dose oral cholera vaccine (OCV) strategies over time due to limited OCV availability [88].

A prevailing norm is maintaining accountability to global polio eradication goals, even in crises [121]. This perspective has led to exceptional measures and concessions to meet these goals, as seen in adopting less desirable vaccination strategies in Afghanistan [33] and securitised responses to vaccination refusal among Afghan refugees in Pakistan [67].

The strong influence of the polio eradication initiative in crisis-affected settings is evident in its ability to impact broader humanitarian policy. This was exemplified by the declaration of a wild poliovirus (WPV) outbreak in Syria and Iraq in 2013, leading to a declaration of a Public Health Emergency of International Concern (PHEIC) by the WHO. This declaration paved the way for an unprecedented cross-border response in Syria involving the WHO, UNICEF, and the GPEI [62]. While the polio eradication program is viewed positively as a source of vaccination infrastructure and qualified personnel for other vaccination efforts [51,74,77,106,116,117,122,123,124,125,126], critics highlight challenges such as operational competition with other interventions and missed opportunities to support routine immunisation [74,78,127]. For example, scheduling overlaps between polio and OCV campaigns in Haiti strained resources and impacted vaccination planning [42].

#### 3.3.4. Governance Arrangements for Designing, Planning, and Providing Vaccination Services (*n* = 53)

The reviewed documents included a moderate amount of information on influential actors involved in designing, planning, and providing vaccination services in countries affected by crises. The WHO, UNICEF, and MOH or equivalent health authorities were the most influential actors. 

The WHO and UNICEF, and to a lesser extent, other UN agencies such as UNHCR and United Nations Relief and Works Agency for Palestine Refugees in the Near East (UNRWA), play significant roles in co-leading vaccination campaigns with MOH [53], actively participating in decision making, for vaccine introduction [128,129] or intervention planning [46,130,131], and managing vaccination data [81,123]. They are sources of trusted information for MOH and other health actors, providing coverage estimates, data, and insights essential for planning and intervention [56,57,63,80,132,133,134]. The WHO and UNICEF play other roles unique to their areas of expertise. For example, UNICEF plays an essential role in operationalising vaccination services through large-scale support to vaccination infrastructure [121]. At the same time, the WHO leverages its social and cultural capital to influence decisions such as vaccine introduction [83], coordination of vaccination efforts in governance vacuums [48,49], and pushing for the use of vaccination in multisector outbreak responses [106].

While collaborating with the WHO and UNICEF, MOH often maintain independence and control over vaccination decisions. They make unilateral decisions, as exemplified by the Haitian MOH’s 2011 decision to conduct an OCV campaign for cholera [42,44], and can assert dominance while relying on external support [64]. MOH may use coercion to enforce vaccination compliance [67] and occasionally override technical advice for political or operational reasons [106,113].

Challenges to MOH authority can arise from organisations like UNHCR [50], national bioethics committees [42], and NGOs [125], leading to disrupted coordination or delays in vaccination responses. Armed anti-government actors in conflict settings can also act as gatekeepers, undermining MOH authority by impeding physical access to targeted populations or the delivery of vaccines [32,33,34,76,77,78,134,135,136,137]. Negotiations with these actors may result in temporary ceasefires [135,138] or access agreements [47].

In-country decision-making spaces fall into two main categories: crisis-specific and longer-term development-related spaces. Crisis-specific spaces involve multi-actor platforms aimed at joint decision making and coordination for emergency vaccination interventions, often used in response to outbreaks or crises [46,65,81,107,131,139,140]. These spaces may vary in their duration and composition. Long-term development planning spaces, such as those related to RED, EPI, or polio eradication, impact vaccination strategies for crisis-affected populations [33,72,74,118]. Furthermore, advisory bodies, such as National Immunization Technical Advisory Groups (NITAGs) and informal consultative groups, also influence vaccination decisions in some countries [36,39].

A prevailing norm is the consensus among stakeholders that there should be a central vaccination authority, mainly the government of a crisis-affected country. This can be observed in the reluctance of some external actors to challenge or appeal government vaccination authorisation decisions [24] and governance adaptations when the government is unable or unwilling to enforce its authority [48,123]. In addition, collaboration is a common culture among vaccination stakeholders, leading to multi-actor campaigns and partnerships to provide vaccination services [42,65,73].

### 3.4. Reported Problems and Solutions Related to Childhood Vaccination Governance in Crisis Settings

Some reviewed documents reported governance challenges and their consequences in vaccination within crisis-affected settings. Six problematic governance features were identified (Table 4). 

Weak in-country governance, encompassing oversight, financial management, and accountability mechanisms, was the most commonly reported issue, reflected in criticism of the polio eradication programs in Afghanistan, Nigeria, and Pakistan [38,85,101,141] and of the management of GAVI funds in DRC [100,120,142]. Solutions implemented or recommended included a shift towards results-based financing [100,120,142], closer coordination between formal health clusters and local health authorities in opposition-controlled areas, such as northern Syria, to enhance service provision and reduce political interference [48], and an accountability dashboard to improve oversight [85].Distrust among vaccination actors, particularly international agencies towards national governments, was observed in multiple instances, such as scepticism by the CDC and the GPEI about Pakistan’s ability to meet polio targets [32], concerns about the government’s expertise and leadership capacity for polio eradication in Afghanistan [33], preconceived notions of corruption and bureaucracy within the Lebanese government during the early years of the Syrian refugee crisis [50], and a lack of confidence by international donors in South Sudan’s MOH capacities [143]. The suggested solutions include motivating governments to implement recommendations [32], adopting a supportive attitude towards governments [33], investing in their capacity [143], and facilitating inclusive discussions to build better partnerships between state and non-state actors [47].Imbalanced financial interdependencies between global and country stakeholders were reported, leading to concerns about sustainability. This was observed in Somalia, where concerns arose over the withdrawal of GPEI funds [51]. Hsu et al. recommended that some external funding should continue in Somalia and that the transition of polio resources should be done effectively and with maximum impact [51]. In Afghanistan, stakeholders were worried about maintaining vaccination services after donor support ended [101]. In Haiti, the dependence on external donor financing for vaccination highlighted the need for innovative health financing and more robust public health sector governance to reduce reliance on external assistance [41].Inflexible policies and bureaucratic procedures sometimes hindered vaccination efforts, with recommendations for more flexibility. These include the requirement for official approval from host governments for vaccination programs carried out by external agencies [24,121,144] and GAVI funding policies for countries that do not meet eligibility criteria [145]. Recommendations were made for MOH in crisis-affected countries to allow campaigns with vaccines not yet included in the national immunisation schedule [24], GAVI policy reforms [145], and simplification of application processes to the OCV stockpile [43,90].Competing accountability streams emerged, such as the conflict between polio eradication and routine immunisation goals [31,101] or when UN agencies and international NGOs have internal mandates and policies that may conflict with those of governments [50]. The authors proposed no specific solutions to address these issues.Blurred roles and responsibilities among vaccination actors were reported, impacting data ownership [125], performance management [64], and reporting systems [35]. Some authors also described instances where the roles and responsibilities of vaccination actors are blurred. The authors proposed no specific solutions to address these issues.

Some authors described the consequences of existing governance problems on vaccination efforts. One consequence is slow or hesitant vaccination responses, manifested as delayed publishing of outbreak response plans [62,106], prolonged vaccine deliberations by actors [42,105,113,146], and delayed implementation of campaigns [18]. It is noted that these delays are not consistent with the aim of timely prevention [18]. Recommendations include devolving decision making to frontline staff for more flexible responses [123]. Other authors associated governance problems with inappropriate vaccination intervention design that does not always align with the actual needs of crisis-affected areas [18], is not informed by guidelines or a framework [73], or does not sufficiently address the root causes [31] or prioritise vaccination [56,90,106]. To address this, suggestions include introducing the WHO’s decision-making framework for vaccination in acute humanitarian emergencies to national and sub-national health authorities [24] and implementing generic multi-antigen mass vaccination campaigns in all acute crises [18]. Finally, one case demonstrated how extensive NGO support for government-led vaccination activities does not guarantee satisfactory program performance or coverage [64].

## 4. Discussion

This is the first paper to review the literature on the features and governance of childhood vaccination services in crisis-affected settings. The review highlights that significant gaps remain despite a moderate body of evidence. Only a few of the reviewed documents directly studied or discussed governance issues, and most findings were constructed by piecing together information retrieved from different documents, guided by the analytical tools of the governance analysis framework (GAF).

The available literature is mainly related to active outbreaks in conflict-affected settings and predominantly to the mass delivery of polio, cholera, and measles vaccines. This finding is consistent with previous research reporting that a limited range of vaccines is used in crisis-affected settings [18,20,21]. We also found there is little systematic reporting of the vaccine delivery strategies used or features of the targeted population, such as displacement status, vaccination status, or age group. A minority of the reviewed documents reported the sources of vaccines, vaccine supplies, cold chain, or funding for these activities. This finding is also consistent with the existing literature on the limited availability of publicly available information on humanitarian and global health funding [147,148,149,150] and a general dearth of published reports of humanitarian health responses, including, for example, reports of reactive or preventive vaccination campaigns in emergency settings [151], or detailed descriptions of infectious disease interventions delivered to women and children in conflict settings [152].

Our findings reveal wide-ranging definitions and interpretations of governance in the context of childhood vaccination in crisis-affected settings. Notably, most interpretations imply some form of vertical authority and regulatory power, with a prominent role for the government of a crisis-affected country. None of the definitions encountered in the reviewed documents acknowledged that the governance of childhood vaccination, like other global health issues, involves formal and informal, vertical and horizontal processes among multiple stakeholders within and outside crisis-affected countries [29,153]. Evidence suggests that the concept of governance in the global health arena is not well understood, and a shared understanding is necessary to inform discussions about the future of governing global health issues [153,154]. This varied understanding may explain some of the findings, such as prioritising some vaccines (e.g., polio) over others (e.g., PCV and rotavirus), due to control over decision making by a few strategic actors. While a unifying definition of governance may not be necessary, an appreciation of the true nature of governance by actors is essential to shed light on opportunities for fully harnessing the resources and capabilities of involved actors that enable new forms of collective action [155]. This will also allow a recognition of the importance of including critical actors, such as crisis-affected communities, who are excluded from current governance structures. 

The reported governance problems in our review indicate that, currently, the governance of childhood vaccination in crisis-affected settings is multi-actor but fragmented, resulting in inappropriate design, timeliness, and performance of vaccination responses. Specifically, we identified that access to vaccine stocks is potentially the most influential factor in designing and planning interventions. We argue that these findings may be partially explained by insufficient incorporation of collaboration, equity, shared strategic vision, and accountability principles—four universal principles of good global governance generally and health specifically [156,157,158,159,160]. 

Our findings suggest an inability to harness the creativity and resources potentially available among actors. This manifests through the inequitable participation of actors in all or some governance structures and the ineffective collaboration between already involved actors. For example, neither crisis-affected populations nor private health sector providers in crisis-affected countries seem involved in deciding which vaccines are offered, how to deliver them, and to whom. This is despite wide acknowledgement that meaningful participation of crisis-affected populations in the design of humanitarian responses is likely to result in timely, appropriate, and effective humanitarian responses [161,162]. Furthermore, although governments of crisis-affected countries generally play a leading role in decision making about vaccination responses within their countries, we identified instances when they are either sidelined or dominated by international actors. This mutual tension and lack of trust between international humanitarian actors and governments of crisis-affected countries has been reported in the literature and is particularly problematic in conflict settings where governments may play a role in obstructing or withholding assistance from crisis-affected populations [163]. We also note that global initiatives to address significant bottlenecks to childhood vaccination, such as improving access to vaccine stocks through global stockpiles, seem to have further consolidated the power and dominance of the largest UN agencies and international organisations, such as UNICEF, the WHO, ICRC, and MSF, not just at global level, but within country responses. Previous evidence suggests that this phenomenon occurs within global humanitarian interagency mechanisms and may result in these agencies setting and defining collective outcomes without meaningful participation by less powerful but equally important actors [163].

Good governance also requires shared commitment amongst all actors across the humanitarian-development spectrum to a strategic vision governed by a set of ideals—this incentivises individual actors to contribute to a shared commitment within their mandates, means, and capacities and serves to unite the individual governance structures and arrangements to serve a common goal [158]. Our findings suggest no shared strategic vision of what actors should strive to achieve through vaccination in crisis settings. For example, we identified competition between polio eradication goals and preventing excess morbidity and mortality from VPDs through vaccination. Researchers have also previously noted the dominance of polio vaccination activities in crisis-affected settings despite a low or moderate severity risk [18,164]. The findings also reveal that the accountability of individual institutions to their own internal mandates and competition between institutional accountabilities may result in delayed and hesitant decisions about vaccination interventions. This tension was also previously identified in healthcare governance during humanitarian responses [23]. Given the diversity of independent actors involved, practising accountability is complex and multipolar [165]. We argue that improving accountability requires first addressing meaningful and equitable participation in governance. This will ensure that accountability mechanisms account for the power and influence of the different actors involved.

This review has important limitations. First, we restricted our review to the published scholarly literature and grey literature. The results from the grey literature informed the description of actors and vaccine interventions but contributed less towards the governance-related findings. This is unsurprising as humanitarian responders rarely undertake formal publishable documentation, and detailed reports are usually for internal use. Therefore, this review may suffer from publication bias.

Secondly, we acknowledge that specific crisis contexts and vaccines dominate the literature. We also restricted eligible literature to reports published only in English, and we may have missed important information from crises that occur in countries where English is not widely used (e.g., Spanish- and French-speaking countries in Africa, Central America and South America).

A third limitation is that controversial or adversarial views are less likely to be published for fear of alienating key actors or losing access to funding or operating footprint. This may mean that specific challenges with vaccination services may have been overlooked or insufficiently emphasised in this review. To complement this review, we have conducted a qualitative study using in-depth individual interviews with various global and national vaccination actors that will be published separately.

### Potential Implications of Findings for Programming and Future Research

This review highlights the need for improved documentation of vaccination interventions in emergencies, particularly where interventions are not in response to an outbreak. We encourage humanitarian actors to document and publish descriptions of their vaccination interventions and the findings of any evaluations they undertake and to document and publish their decision-making processes to promote transparency and foster accountability. We also encourage donors to incentivise the publishing of vaccination interventions that they fund.

The limited information on how vaccination decisions are made and how intervention design and planning are governed indicates the need for rigorous empirical research within decision-making spaces and amongst global and national vaccination stakeholders across the humanitarian-development spectrum. This will help elucidate the nuances of stakeholder influence and interactions and the implications on the timeliness and appropriateness of childhood vaccination interventions. It will also help investigate how reported governance problems are interconnected or may compound each other in crisis-affected settings. More importantly, research that elicits the perspectives of crisis-affected communities and other excluded actors on their current and desired participation in the design and planning of childhood vaccination interventions is paramount.

There is much variation in the conceptualisations and interpretations of governance across the literature, indicating the need for a shared understanding of the current and desired state of childhood vaccination governance in crisis-affected settings. While further empirical research can shed light on the current governance status, consensus-building exercises amongst stakeholders to agree on a shared strategic vision, measures for equitable participation in governance structures, and associated accountability mechanisms are necessary. These exercises should culminate in a voluntary alliance around a common goal, and common norms and standards, similar to the Core Humanitarian Standard (CHS) [162], and an independent audit mechanism for the collective performance of the system, such as the Humanitarian Quality Assurance Initiative (HQAI) [166], which is linked to conditional funding.

## 5. Conclusions

This review highlights the need for improved documentation of vaccination interventions in emergencies. There is limited information available on how vaccination decisions are made and how intervention design and planning is governed. There is a need for a shared understanding among vaccination actors of the current status and desired state of governance of childhood vaccination in crisis-affected settings and for empirical research within decision-making spaces across the humanitarian-development spectrum.

## Figures and Tables

**Figure 1 vaccines-11-01853-f001:**
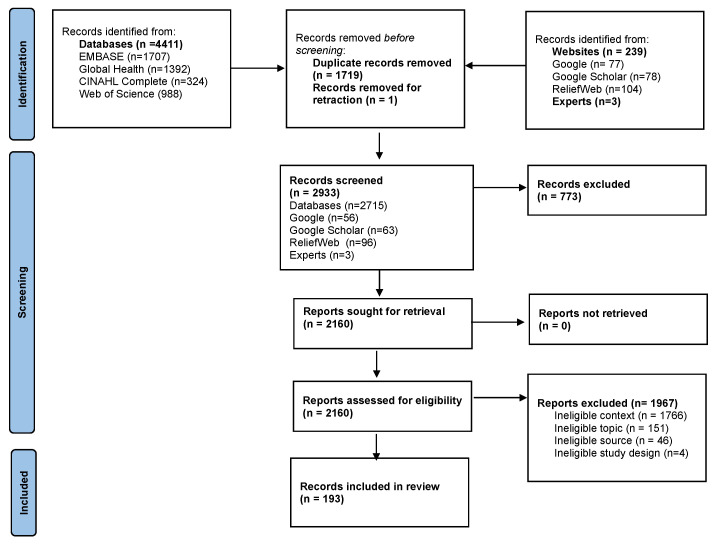
Results of record screening and selection process.

**Figure 2 vaccines-11-01853-f002:**
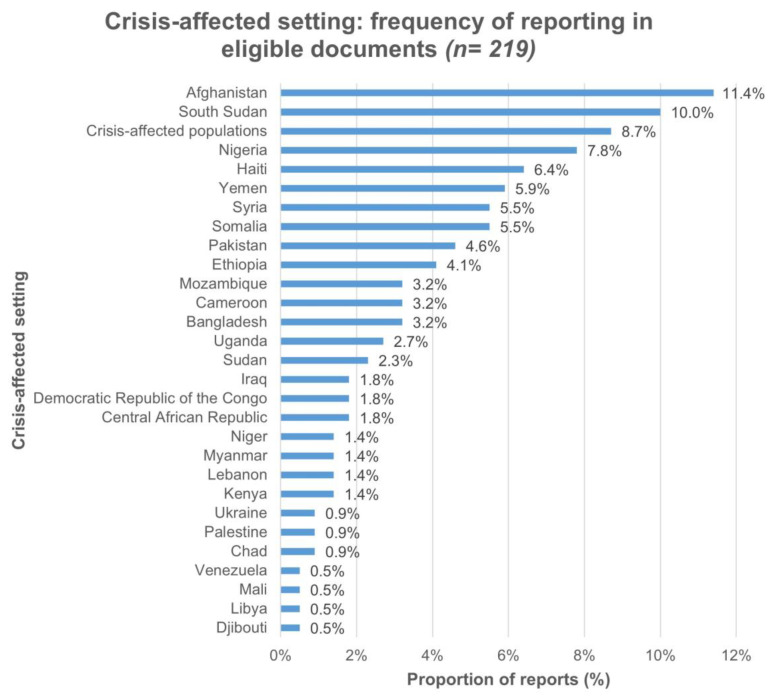
Crisis-affected setting reports in 193 included documents.

**Figure 3 vaccines-11-01853-f003:**
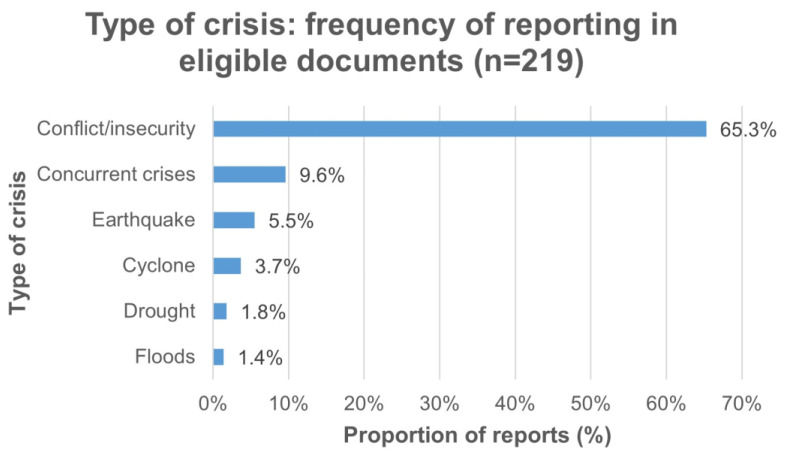
Crisis typology reports in 193 included documents.

**Figure 4 vaccines-11-01853-f004:**
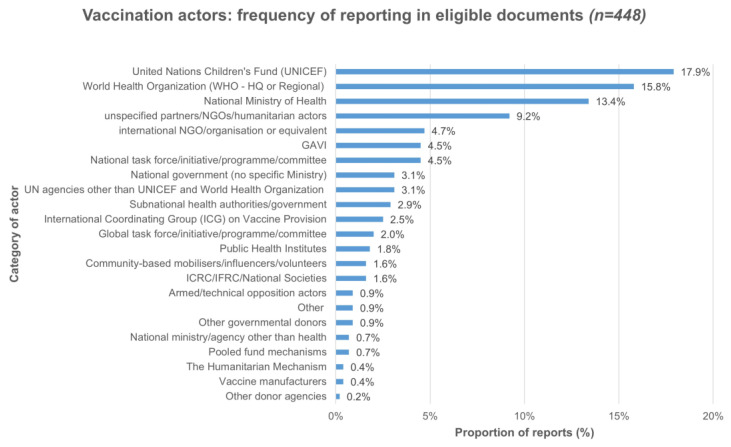
Vaccination actor reports in 193 included documents.

**Figure 5 vaccines-11-01853-f005:**
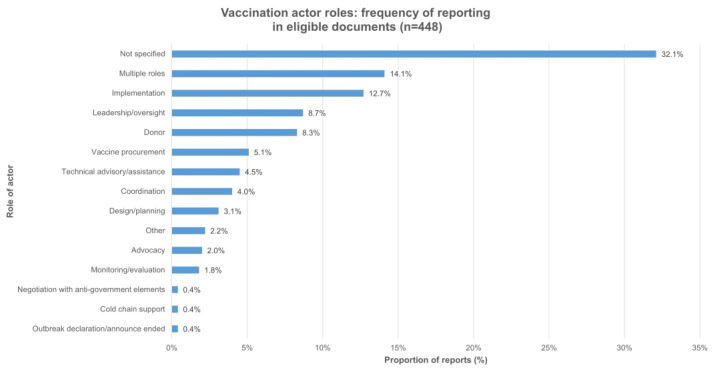
Vaccination actor role reports in 193 included documents.

**Table 1 vaccines-11-01853-t001:** Eligibility criteria.

Criterion	Inclusion	Exclusion
Context and population	The document refers to one or more crises with a consolidated appeal/humanitarian response plan during 2010–2021 (Appendix A)	Studies of people originally from one or more of the eligible crises but now residing outside of the crisis-affected region (e.g., refugees and migrants)
Topic	1. Vaccination: The document describes any services that aim to deliver antigens through routine or supplementary modalities to a population that includes children under five years of age AND 2a. Governance: (i) Documents with the terms ‘governance’ or ‘accountability’ or ‘leadership’ or ‘stewardship’ or ‘decision-making’ in the title or abstract/executive summary, OR (ii) Documents that describe any mechanisms of control, compliance, or performance assurance OR 2b. Vaccine service planning: Documents that describe one or more of the following features of vaccine services: actors involved, vaccine choices, vaccine delivery modalities, vaccine target population, sources/procurement mechanisms of vaccines and cold chain, sources of funding for vaccine services	Vaccination: Antigens not delivered to children under five years of age in crisis-affected settings (e.g., COVID-19)
Source type	• Narrative reports: annual, technical, financial, operational • Guidance documents • Research articles • Commentaries/editorials, letters to the editor in peer-reviewed journals • Funding databases • Literature reviews • Conference abstracts • Strategy and policy documents/briefs	• Opinion pieces, including speeches and blog posts • Book chapters • Audio/video reports • Conference abstracts covering the same material as a full-text publication • Social media/media articles • Legal documents • Inaccessible full-text documents
Study design	All primary, observational, mixed method, quantitative or qualitative study designs, including case studies Literature reviews Any grey literature	Interventional studies
Publication date	1 January 2010–31 December 2021	
Language	English	

**Table 2 vaccines-11-01853-t002:** Characteristics of vaccine reports in 193 included documents.

Characteristic		Number of Reports (*n* = 219)	Proportion of All Reports
Vaccine	DTP-containing vaccines	12	4.0%
	Measles/measles-containing vaccine	54	18.2%
	Rubella	9	3.0%
	Polio (OPV or IPV)	89	30.0%
	Meningococcal meningitis	4	1.3%
	Yellow fever	3	1.0%
	Hepatitis A/E/B/G	1	0.3%
	Cholera	54	18.2%
	Pneumococcal conjugate vaccine	5	1.7%
	Rotavirus	3	1.0%
	Routine vaccines	51	17.2%
	Other	3	1.0%
	Not specified	9	3.0%
Infection transmission scenarios	Active epidemic/transmission	128	43.1%
	No active epidemic/transmission	53	17.8%
	Not specified	116	39.1%
Aims of vaccination	Epidemic prevention	35	11.8%
	Epidemic response	86	29.0%
	Increase routine coverage	62	20.9%
	Disease elimination/eradication	43	14.5%
	Other	4	1.3%
	Not specified	67	22.6%
Modalities of vaccine delivery	Routine	38	12.8%
	Supplementary	32	10.8%
	Mass	120	40.4%
	Combined/multiple	33	11.1%
	Not specified	74	24.9%
	Routine	38	12.8%
Approaches for vaccine delivery	Fixed	9	3.0%
	Mobile	5	1.7%
	Outreach	2	0.7%
	House-to-house	6	2.0%
	Transit point teams	6	2.0%
	Combined approach	45	15.2%
	Other	10	3.4%
	Not specified	214	72.1%
Typology of the targeted population	Refugee	24	8.1%
	Returnee	3	1.0%
	Internally displaced persons (IDP)	28	9.4%
	Transiting/migrant	2	0.7%
	Crisis-affected non-displaced	5	1.7%
	Low-coverage/high-risk areas	47	15.8%
	Zero-dose/unvaccinated	2	0.7%
	Multiple	65	21.9%
	Other	11	3.7%
	Not specified	110	37.0%
The age group of the targeted population	Under 1	13	4.4%
	Under 2	2	0.7%
	Under 5	60	20.2%
	Under 15	26	8.8%
	Other	36	12.1%
	Not specified	160	53.9%
Source of vaccine	Vaccine manufacturers	2	0.7%
	Global Polio Eradication Initiative (GPEI)	1	0.3%
	GAVI	2	0.7%
	International Coordinating Group (ICG) on Vaccine Provision	20	6.7%
	UNICEF	3	1.0%
	UNICEF + international NGO/organisation or equivalent	1	0.3%
	The Humanitarian Mechanism	1	0.3%
	Not specified	267	89.9%
Source of funding for vaccines	Pooled fund mechanisms	2	0.7%
	GAVI	16	5.4%
	Other governmental donors	2	0.7%
	International NGO/organisation or equivalent	3	1.0%
	International Coordinating Group (ICG) on Vaccine Provision	2	0.7%
	UNICEF	2	0.7%
	WHO + International NGO/organisation or equivalent	1	0.3%
	Vaccine manufacturers	1	0.3%
	WHO + ministry of health	1	0.3%
	WHO + UNICEF + ministry of health	1	0.3%
	UNICEF + International NGO/organisation or equivalent	1	0.3%
	Global Polio Eradication Initiative (GPEI)	1	0.3%
	Not specified	264	88.9%
Cost of vaccine	Less than USD 3 per dose	4	1.3%
	More than USD 3 per dose	2	0.7%
	Not specified	291	98.0%
Source of cold chain	Ministry of health	1	0.5%
	UNICEF	3	1.6%
	WHO	1	0.5%
	Not specified	188	97.4%
Cost of cold chain (per dose, USD)	0.35	1	0.5%
	Not specified	192	99.5%
Source of vaccination supplies	Ministry of health	1	0.5%
	GPEI	1	0.5%
	UNICEF	4	2.1%
	Not specified	187	96.9%
Source of funding for vaccination supplies	GAVI	1	0.5%
	ECHO	1	0.5%
	UNICEF	2	1.0%
	WHO + UNICEF	1	0.5%
	Not specified	188	97.4%
Source of funding for vaccination service delivery	GAVI	4	2.1%
	UNICEF	1	0.5%
	Subnational health authorities/government	1	0.5%
	GPEI	1	0.5%
	International NGO/organisation or equivalent	2	1.0%
	Ministry of health	1	0.5%
	ECHO	1	0.5%
	Not specified	182	94.3%
Cost of vaccination service delivery (USD per dose)	0.60–0.69	2	1.1%
	0.70–0.79	1	0.5%
	0.80–0.89	1	0.5%
	Not specified	186	97.9%

**Table 3 vaccines-11-01853-t003:** Definitions and interpretations of the governance of childhood vaccination in crisis-affected settings in included documents (*n* = 66).

Concept	Interpretations
Programme Management	The existence/quality of frameworks of authority and accountability that define or control the management/performance of vaccination programmes, including: Policies, e.g., policies that seek out and vaccinate refugees and migrants Systems or processes, e.g., coherent national systems that review/address operational immunisation issues such as hesitancy, staff performance, etc.—program structures that define clear roles and responsibilities of actors—periodic reviews of staff performance and capacities Strategies or plans, e.g., to improve program management and performance Protocols or guidelines, e.g., to guide/inform vaccination activities—for adverse events following immunisation (AEFI) reporting/notification systems
Leadership and Ownership	• Commitment by senior government officials • Command and control/coordination of diverse actors • Developing and implementing a vaccination intervention • Ability of national governments to obtain the cooperation of the population in achieving vaccination aims • Declaring or reporting outbreaks • Overarching vaccination strategy is appropriate to the context
Accountability	• Measure and report on program outcomes, e.g., administrative vaccination coverage, post-vaccination coverage surveys, using independent monitors, using outbreak/AEFI reporting/notification systems • Implement recommendations of ‘experts’ to meet global targets, e.g., external advisory groups • Leverage the need to meet global targets to justify proposed/implemented vaccination activities • Take measures to improve program performance • Develop or use accountability frameworks or similar • Involve affected/target population—end users
Decision making	Framed as ‘evidence-based’: is/should be informed by data/information specific to crisis/response, or credible frameworks/guidelines, or scientific research Framed as ‘prioritisation’ of target populations/geographical areas for vaccination or of vaccination over other public health interventions, or specific vaccines over others
Other definitions of governance	• Sustained financial and technical independence of actors to define and achieve vaccination aims • Official recognition of health workers and health systems delivering vaccination services as legitimate

**Table 4 vaccines-11-01853-t004:** Reported problems and solutions related to childhood vaccination governance in crisis settings.

Reported Problems	Reported Solutions
Weak in-country governance, e.g., • inadequate oversight of the quality of vaccination activities • lack of reliable financial and accountability mechanisms • lack of devolved vaccination activity coordination bodies • inadequate management of funds	Accountability tools, such as a dashboard Results-based financing and legal reform of financial management Closer coordination between formal health coordination bodies and de facto health authorities in governance vacuums
Distrust between vaccination actors, e.g., • by international actors towards national governments in crisis-affected countries • between state and non-state in-country actors	Ensuring governments are motivated to implement recommendations Supportive attitudes by international actors towards governments in crisis-affected countries International investment in the capacity of governments and national NGOs International actors can facilitate inclusive discussions between state and non-state actors
Imbalanced financial interdependencies between global and country vaccination stakeholders, e.g., • chronic reliance on external aid for vaccination	Crisis-affected countries should explore internal revenues and novel health financing solutions to sustain services Preservice education and training and robust governance capacity within the public health sector
Inflexible policies or bureaucratic procedures, e.g., • restrictive GAVI funding policies • host government authorisation for vaccination interventions • restrictive procedures to access vaccine stockpiles	Further reform of GAVI funding policies in non-eligible crisis-affected countries and eligible countries facing complex emergencies Ministries of health in crisis-affected countries should adopt flexible policies for vaccination in emergencies Simplifications of mechanisms and application processes to access the global OCV stockpiles
Competing accountability streams, e.g., • competition between polio eradication and EPI goals, • competing institutional accountabilities between external actors and governments in crisis-affected countries	None
Blurred responsibilities in multi-actor vaccination interventions, e.g., • ownership of vaccination data, • responsibility for poor vaccination programme performance, • parallel adverse events from immunisation (AEFI) reporting systems	None
Vaccination interventions are hesitant or slow, e.g., • delayed publication of outbreak response plans • prolonged deliberation to decide on the use of vaccines during outbreaks • delayed implementation of reactive campaigns • lack of prioritisation of vaccination over other interventions	Devolvement of decision making to frontline staff
Inappropriate vaccination intervention design, e.g., • Inappropriate vaccines • Inappropriate vaccine delivery strategies	Proactive introduction of the decision-making framework for vaccination in acute humanitarian emergencies to health authorities Generic multi-antigen mass campaigns of the main VPDs
Extensive support to health authorities does not guarantee satisfactory vaccination coverage and program performance	None

## Data Availability

All data generated or analyzed during this study are included in this published article and its Appendix A.

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
