# Peer review of "The Governance of Childhood Vaccination Services in Crisis Settings: A Scoping Review"

_vaccines, 2023, doi:10.3390/vaccines11121853_

Round 1
Reviewer 1 Report
Comments and Suggestions for Authors
The study "The Governance of Childhood Vaccination Services in Crisis 2 Settings: A Scoping Review" aims to investigate the governance of childhood vaccination services in crisis-affected settings. It employs a comprehensive scoping review, analyzing various documents and literature. Key findings reveal that governance in these settings is multi-actor but fragmented, leading to challenges in the design and implementation of vaccination programs. The study also uncovers a lack of systematic reporting on vaccine delivery strategies and targeted populations. Notably, it highlights the need for a shared strategic vision and greater inclusion of local communities in decision-making processes to enhance the effectiveness of vaccination interventions in crisis situations. Although the paper presents a certain scientific interest, there are some concerns regarding the validity of the presented data and the overall results. Here are some important comments:
1. In the Introduction section, please emphasize the broader impact of challenges in vaccine-preventable diseases in crisis settings on global health security and development goals.
2. Please, discuss more thoroughly the gaps in current research, particularly regarding the governance of vaccination services in crisis settings, to strengthen the study's rationale.
3. Please, elaborate on how socio-economic and gender disparities specifically impact vaccine service delivery and uptake in crisis settings.
5. Provide a critical analysis of innovations and strategies mentioned in the introduction, such as microarray patches and Gavi's ZIP, in crisis settings.
6. Analyze the effectiveness and limitations of the WHO decision-making framework in the context of crisis settings.
7. In the Materials and Methods section, please Justify the choice of Arksey and O’Malley’s scoping framework and its revisions and refinements for this particular study.
8. Explore how varied understandings of governance might impact the implementation of vaccination programs in crisis settings.
9. The presentation of data from eligible documents is thorough. However, consider using visual aids like charts or graphs to enhance the readability and understanding of the data, especially where large quantities of data are presented (e.g., breakdown of crisis settings, vaccine types, and actors involved).
10. The detailed exploration of governance arrangements across various domains (funding, access to vaccine stocks, setting goals and standards, and service provision) is commendable. It would be useful to highlight any emerging patterns or common challenges across these domains to provide a more cohesive analysis.
11. While the document covers a range of actors, the perspectives of local communities and end-users of these vaccination services seem underrepresented. Consider discussing the importance of including these perspectives in understanding and improving vaccination governance.
12. The section on governance challenges and solutions is insightful. Expanding this section to include more on how these challenges are interconnected and how they might compound one another in crisis settings could deepen the analysis.
13. Given the fast-paced changes in global health, particularly in crisis settings, consider whether more recent developments or literature (post-May 2022) might impact your findings.
14. Please, discuss strategies or models to effectively include crisis-affected communities in decision-making processes.
15. The language used is appropriate for an academic paper, but consider simplifying complex sentences for better readability, especially for a diverse audience that might include non-specialists.
16. While the paper presents a detailed scoping review, suggesting areas for future research based on identified gaps or emerging questions would be valuable.
Comments on the Quality of English LanguageMinor editing of English language required
Reviewer 2 Report
Comments and Suggestions for Authors
The manuscript reviews the vaccine development, delivery and governance policies in developing and underdeveloped countries. The review is very well written and comprehensive. I have a few minor comments:
1) The review is with regards to vaccines for children. How has the results changed since the COVID-19 pandemic? Were the identified problems better handled in the recent pandemic?
2) Among the sources used in this review, is there is preference on documents available from vaccine manufacturers over those available in research papers or review articles? Was there a preference for local vaccine manufacturers, local government policies, over global vaccine companies in decision-making?
3) The cost, accessibility and delivery infrastructure for vaccines is an important factor as the authors have identified. The pandemic has exposed some of these problems. Continuing with my first point, some text about the accessibility and mass delivery of vaccines could be included. A review paper covers the cost and accessibility of COVID-19 technologies: Benda A, et al, COVID-19 Testing and Diagnostics: A Review of Commercialized Technologies for Cost, Convenience and Quality of Tests. Sensors. 2021, 21(19):6581.
4) In Table 6, the suggested solutions are by your team or those suggested in the sources? It would be good to mention this point in the Table. The last point in Table 6 has a grammar mistake "Extensive support to does not guarantee satisfactory vaccination coverage and program performance"
